# Clinical Management of Intraosseous Access in Adults in Critical Situations for Health Professionals

**DOI:** 10.3390/healthcare10020367

**Published:** 2022-02-14

**Authors:** Álvaro Astasio-Picado, Paula Cobos-Moreno, Beatriz Gómez-Martín, María del Carmen Zabala-Baños, Claudia Aranda-Martín

**Affiliations:** 1Nursing and Physiotherapy Department, Faculty of Health Sciences, University of Castilla-La Mancha, Talavera de la Reina (Toledo), 45600 Toledo, Spain; carmen.zabala@uclm.es (M.d.C.Z.-B.); claudia.aranda@alu.uclm.es (C.A.-M.); 2Nursing Department, University of Extremadura, Plasencia (Cáceres), 10600 Plasencia, Spain; pacobosm@unex.es (P.C.-M.); bgm@unex.es (B.G.-M.)

**Keywords:** intraosseous access, emergencies, adult

## Abstract

There are health professionals who are unaware of the ideal management of the intraosseous route, despite the fact that it has been scientifically considered an alternative to the peripheral venous route when the patient is in critical condition. Thanks to continuous development, there has been a need to provide emergency services with materials that manage to provide satisfactory care, despite the difficulties faced by health personnel. Objectives: The objective of this systematic bibliographic review is to update the theoretical and practical knowledge and strategies for the insertion and proper management of the intraosseous route as an emergency vascular access for nursing professionals. Data sources, study eligibility criteria: The search for the articles was carried out in various scientific databases with the help of a search string (January 2015 and May 2021), which combined the keywords and Boolean operators. Study appraisal and synthesis methods: Eighteen articles were chosen after a review of 1920 database articles, following the application of the inclusion and exclusion criteria. Results: Intraosseous infusion is an effective and safe technique, which increases patient survival. Therefore, it is of crucial importance that all nursing professionals know how to handle the different intraosseous devices in situations in which it is not possible to achieve immediate peripheral venous access. Conclusions and implications of key findings: It is of great need to have devices or fast and effective alternatives that allow us to develop safe interventions by health professionals.

## 1. Introduction

Intraosseous injection (IO) is a technique indicated in life-threatening situations for both adult and pediatric patients, in which the infusion of medications or liquids cannot be delayed in time, and due to age or circulatory collapse, vascular access cannot be achieved in about 60–90 s, with a maximum of up to three puncture attempts. Above all, it is used in patients in a state of shock who need to be administered blood, crystalloids, or colloids in situations of anaphylaxis, severe burns, obesity, status epilepticus, trapped, dehydrated, multiple trauma, altered level of consciousness, etc. It is an effective technique and easily accessible, with minimal complications at the time of insertion [1]. The American Heart Association (AHA) and the European Resuscitation Council (ERC) of 2015 recommended the intraosseous route after the peripheral route and prior to the central venous route, in cases of difficulty or delay in venous cannulation [1,2]. On the part of nurses, it is essential to carry out rapid action, in order to reduce the probability of morbidity and mortality and favor the recovery of the patient [1,2,3,4]. Nursing professionals can perform intraosseous insertion since this technique was approved and is reflected in the NIC code 2303 “Administration of intraosseous medication” by inserting a needle through the bone with the aim of administering fluids, blood, or medications [1].

Anatomically, the main support of the human body is the skeletal system. This is responsible for the protection of internal organs, locomotion, support, and stabilization. We can classify bones into four different categories: long (e.g., the tibia and humerus), short (e.g., the phalanges), flat (e.g., the sternum), and irregular (e.g., the vertebrae) [1,4]. The bone is divided into three main parts: the epiphysis, the diaphysis, and the process. The epiphysis is made up of spongy tissue in the center where the red marrow is located, which contains the blood cells responsible for developing the body’s immune defenses. The metaphysis is the junction area of the epiphysis and the diaphysis and in it is the growth plate. The diaphysis is the internal part of the bone and in it we can find the yellow marrow formed by lipids. The apophysis is the protruding part of the bone that communicates with another bone or muscle, the endosteum forms the inner thin membranous lining of the bone cavity, and the periosteum is the membrane that covers the external face of the bone and the internal osteogenic layer [1,4].

The medullary cavity of the long bones of the human body is a network rich in sinusoid capillaries that allows the access of drugs and fluids to the blood circulation, with a speed similar to that of a peripheral venous access, in which they drain into a large central venous sinus that communicates with the general venous circulation through the emissary and nutritional veins. This venous sinus does not collapse even in situations in which the patient is in cardiorespiratory arrest due to the presence of spicules in the medullary cavity and the hardness of the compact bone [4]. The infusion rate will vary depending on the device used, the caliber, the chosen puncture site, and the external pressure exerted [4].

Physiologically, the bone begins to grow and ossify from the epiphysis to the diaphysis. Between them is the epiphyseal or growth plate that will stop growing when said plate turns into bone. Over time, the bone marrow becomes less vascularized yellow bone marrow. Therefore, the insertion sites are not the same in adult and pediatric patients. Generally, IO vascular access catheters are placed at the proximal and distal ends of long bones, because compact bone is thinner and has a greater abundance of cancellous bone [5,6]. Anatomically, in the medullary space of the bone we can find red and yellow bone marrow, apart from a hypercoagulable fibrin mesh. Between them, they form a thick substance that offers resistance to the infusion of blood. Therefore, it is necessary to perform a rapid wash with physiological saline at 9% and begin with the administration of medications and fluids. Depending on the insertion site, fluids drain into the proximal humerus and drains into the axillary vein, the proximal tibia drains into the popliteal vein, the distal tibia drains into the greater saphenous vein, and the sternum drains into the internal mammary veins and azygos [6,7,8,9]. Before channeling an intraosseous route, we must pay attention to the age of the patient and the pathology, always remembering the contraindications. The patient has the right to know what we are going to do and what to expect, in relation to pain, noise caused by the devices, or other sensations [10,11,12,13,14].

The anatomical sites of insertion in adults are the proximal humerus, distal and proximal tibia, sternum, and iliac crest. The proximal humerus is the first choice in adults, provided there are clear anatomical landmarks. Recent studies reveal that with this location, we can achieve higher infusion rates, less pain on insertion, and greater drug bioavailability. We can achieve flows of up to 5 L/h, since medications and fluids reach the right atrium in just 3 s [10,11,12,13,14]. In the proximal tibia, we can achieve fluid volumes of 1 L/h under pressure. The patient’s leg should be extended, and the insertion point is located 3 cm, about 2 finger widths, just below the patella, and about 2 cm medial along the flat aspect of the tibia [12]. At the distal tibia, the patient’s leg should be extended, and the insertion point is 1–2 cm proximal to the base of the medial malleolus at its midline (3 cm above the crest of the malleolus). [12]. The sternal access allows us to reach the central venous circulation due to its proximity through the mammary glands. The advantage of this access is that it reaches a maximum concentration in the blood with a time similar to that of drugs injected through a central line. The EZ IO T.A.L.O.N device is designed to be placed on the sternum. It is not indicated if the patient requires CPR, since cardiac massage cannot be performed [10,11,12,13,14]. For insertion in the iliac crest, we will place the patient in lateral decubitus to achieve better access. The puncture site is on the inferior aspect of the dorso-iliac spine [13].

Several studies understand with respect to the different IO access devices, that manual intraosseous needles were the first to be used. Among them, the most used are the Jamshidi/Illinois needle, the Sur-Fast needle, and the Dieckmannnn. They are the simplest and cheapest [1].

The Cook Dieckmannn intraosseous needle is composed of a thick metal needle in the shape of a pyramid for adults and a pencil-shaped needle for children and infants (Figure 1). Apart from the needle, it has a sear and a handle, which allows us to exert force on it to cross the cortex of the bone and access its interior. This needle has a black mark located 1 cm from the tip of the catheter, which will be the visual reference point. The needle is inserted perpendicular to the joint. The insertion of this type of needle is carried out by pressure, with a slight rotational movement to penetrate the cortex of the adult patient [1,2,3].

Among the devices for placement by firing, we can distinguish those intended for sternal access (FAST) and those that are not (BIG) [1,2]. The FAST I device (First Access for Shock and Trauma) creates a channel that allows the administration of fluids through the sternum (Figure 2). It is a set of needles around a central unit that is attached to a Luer-Lock type connector. The kit contains an introducer, a line extension system, an adhesive patch, a protective dome, and a dressing to disinfect the skin [1].

The intraosseous infusion gun, or Bone Injection Gun (BIG), is for single use (Figure 3). It consists of a spring, a trigger that fires the pre-assembled catheter, and a compact system with a safety pin. This device can be used in hard cortical bones, but the length of the needle can be conditioned due to the variability of the thickness of the chosen puncture site [1].

The main advantages are the ease of learning the technique, the speed in channeling a vascular access (less than 1 min), the high success rate (>90%) in trained personnel, the guarantee it offers for the administration of fluids and drugs, and the safety it shows as it is a non-collapsible route, especially in situations of cardiorespiratory arrest or shock [11,13]. Furthermore, there is the possibility of obtaining a blood sample through the IO and the ease in recognizing the anatomical landmarks that serve as a guide to locate the puncture site [14,15]. Any drug or solution that can be administered intravenously can be administered intraosseously in the same dose or quantity [16,17]. It has been scientifically proven that the serum levels and efficacy are equivalent to those achieved via the peripheral or central route in both adults and children [11,12,13,14,15,16,17,18].

The general objective of this work is to update the theoretical and practical strategy of the insertion and proper management of the intraosseous route as an emergency vascular access for health professionals.

## 2. Materials and Methods

The preparation of this work was carried out through a systematic bibliographic review of the articles found by searching the following databases: Medline/Pubmed, Research Care, Elsevier/Embase, Scopus, and Google Scholar. To find the best possible scientific evidence, a series of inclusion and exclusion criteria were applied.

The keywords for this review are: intraosseous access; emergencies; and adult. To carry out the bibliographic search, different keywords in English were used, such as: “intraosseous access”, “emergencies”, “intraosseous access”, and “adult”. These have been validated by the DeCS and MeSH. Once selected, the corresponding Boolean operators were used: AND/OR, as well as the necessary parentheses and quotation marks. The final search string is as follows: (“Intraosseous Access”) AND (“adult”). The criteria that were taken into account for the selection of the relevant studies were the following. Inclusion criteria: the period between 2015 and 2021; article type: article review and article research; field: nursing; English language; sample in adult population; and studies that provide scientific evidence justified by the level of indexing of articles in journals according to the latest certainties. Exclusion criteria: articles prior to 2015; language: not English; studies in which the population was minors; studies that do not provide scientific evidence justified by the level of indexing of articles in journals according to the latest certainties.

For the methodological evaluation of the individual studies and the detection of possible biases, the evaluation was carried out using the PEDro Evaluation Scale. This scale consists of 11 items, providing one point for each element that is fulfilled. The articles that obtained a score of 9–10 points have an excellent quality, those between 6 and 8 points have a good quality, those that obtained 4–5 points have an intermediate quality, and, finally, those articles that obtained less than 4 points have a poor methodological quality article [19].

The Scottish Intercollegiate Guidelines Network classification was used in the data analysis and assessment of the levels of evidence, which focused on the quantitative analysis of systematic reviews and the reduction of systematic error. Although it took into account the quality of the methodology, it did not assess the scientific or technological reality of the recommendations [20].

## 3. Results

The research question was constructed following the PICO format (Population/patient, Intervention, Comparator, and Outcomes/Outcomes). Detailed as in adult patients (P), intraosseous access (I) is a better option compared to intravenous access for the administration of fluids and medications in critical situations (C), in order to achieve greater survival and effectiveness (0) (Figure 1).

Below is a table that shows the search strategy used to select the 18 articles selected from the 6 databases, following the criteria of identified studies, duplicate studies, title, abstract, full text, and valid studies of a definitive nature (Table 1). The total number of valid articles is summarized in Appendix A.

According to detailed studies, to determine effective nursing health activities in the use of the intraosseous route, nursing professionals can administer intraosseous medication according to IAS 2303 [1,2,3,4]. In the management of health care, it is essential to carry out a rapid intervention, in order to reduce the probability of morbidity and mortality and favor the recovery of the patient [3,4]. The appropriate intraosseous device must be chosen for the insertion site and the patient’s situation [5,6,7,8]. To describe the benefits and harms of using an intraosseous access compared to venous access, [2,4] both the localization and insertion techniques of the IO are simple [5,6,7]. The percentage of complications is minimal (<1%) [8,9,10,11]. Any drug or solution that can be administered intravenously can be administered intraosseously, in the same dose or quantity [11,12]. IO offers faster cannulation of a vascular access and a higher percentage of successes in trained personnel [13,14,15]. A disadvantage is the high costs of needles and intraosseous games [15,16,17]. According to studies, in order to assess the knowledge and experience that healthcare professionals have regarding the use of the intraosseous route, at present, it continues to be an underused route in emergency situations, despite all the good possibilities it has [17,18]. A comprehensive training and education program, with regularly updated sessions, should allow for rapid and reliable placement of the IO device with proper management and monitoring [18].

## 4. Discussion

Faminu F. shows that the absence of immediate venous access can lead to increased morbidity and mortality in patients [1,6]. Authors such as Janneth R., argue that IO should only be used in situations involving a vital emergency and for a limited time, when venous access is inaccessible and it is a temporary measure until a venous line is achieved [6,11]. Some indicated situations in which an intraosseous route is acceptable are severe bleeding, cardiorespiratory arrest, dehydration, shock, trapped, multiple traumas, severe edema, hypovolemia, sepsis, severe burns, and poisoning, among others. In cases in which the patient is conscious, local anesthesia will be used in the puncture site [1,6,11].

Méndez García J.L., Garagatti Oliveira C., and Janneth R. endorse that the contraindications of the intraosseous route are situations in which this technique should be avoided due to the high probability of complications [4,11]. To do this, we must assess whether the benefits outweigh the risks before acting [6,7]. Therefore, the IO route cannot be used in fractured or punctured bones. In the case of suspected proximal and distal tibia fracture, it is completely contraindicated as they share a common pathway within the bone. On the other hand, if the femur is fractured but the tibia is not, the insertion in the tibia can be performed. Furthermore, we cannot use IO in bones with osteoporosis, osteogenesis, with a history of surgery or prostheses, cellulitis, areas with necrotic tissue or burns, or areas with bone tumors or osteomyelitis. In addition, the aforementioned authors confirm that IO in the lower limb extremities is totally contraindicated in patients with severe abdominal trauma and previously punctured bones [4,9,11]. Distal femur IO is feasible and associated with similar measured performance parameters to other IO sites in adult out-of-hospital cardiac resuscitation for basic and advanced life support personnel [21].

Rodil Díaz J.A. and Taboada Martínez M.L. have shown with a success of 88–95% that the proximal and distal tibia are easily accessible areas, because there is not much fat between the skin and periosteum, becoming a good option for obese patients [1]. Tan et al., observed in a study of 42 patients, 20 with access in the proximal tibia and 22 with access in the distal tibia, that the flow rate of the saline solution administered was higher in the proximal tibia compared to the distal one [7], while Pasley et al., confirmed that the flow velocity in the sternal route was 1.6 times higher than the humeral one, and 3.1 times higher than the tibial one [8].

Montez D.F. et al., in a study of healthy adult volunteers, injected contrast medium through the proximal humerus and captured with fluoroscopy as it entered the heart. The mean time to reach the contrast in the superior vena cava and the right atrium was 2.42 s [9].

According to Petitpas F. et al., complications are infrequent (<1%), since the main problems are due to the lack of experience of the healthcare personnel who perform the technique. The most frequent are due to fluid extravasation around the puncture site, bone fracture, growth plate injury, bone infection, appearance of compartment syndrome when crossing two cortices, or that the bone marrow and fibrin mesh have decreased flow [15]. Petitpas F., et al., also assures that intraosseous insertion should be limited to a few hours until venous access is achieved, without exceeding 24 h. If, after 24 h, we still need to use the intraosseous device, we must insert another in a different area and remove the previous one, since fluids after 24 h may leak. In the cases of infiltration, we will be forced to interrupt the infusion, remove the device, and perform the procedure in another area [15].

Afzali M. et al., include that teaching intraosseous access insertion to students in cadaver courses is an excellent way to introduce early training skills. Additionally, allowing sufficient time for the repetition of the technique leads to an improvement in safety. Therefore, adding these activities in the curricula would increase the competences of each of the students. It is concluded that by attending the IO insertion workshops, a greater degree of confidence is gained than by observing the technique through videos or lectures [18]. Manrique Martínez et al., understand with respect to the different IO access devices, that manual intraosseous needles were the first to be used. Among them, the most used are the Jamshidi/Illinois needle, the Sur-Fast needle, and the Dieckmannnn. They are the simplest and cheapest. Among the devices for placement by firing, we can distinguish those intended for sternal access (FAST) and those that are not (BIG). Fast devices are discouraged when they may interfere with emergency cricothyrotomy or resuscitation maneuvers. Instead, they are very useful in multi-casualty accidents [13].

Median insertion time with EZ-IO was 15 s compared to 20 s with the FAST-Responder. Insertion complications recorded with the EZ-IO included extravasation, aspiration failure, and insertion time >30 s. With the use of the FAST-Responder, complications such as user failure (12.5%) and insertion time > 30 s (12.5%) were reported. Regarding flow, we found that 35.1% of EZ-IO inserts experienced poor flow and required a pressure bag. With the FAST-Responder, flow was reported as very good or good in 85.7% and no inserts had poor flow [18,21,22] (Table 2).

According to the evidence documented by Montez D. et al., motorized drill systems are made of non-porous, fluid resistant plastic. Whenever we are not going to use the drill, we have to place the trigger guard to avoid total consumption of the battery and a discharge. The red light will indicate that there is 10% battery left. They are not designed to be opened, so the batteries are not replaceable [12]. Manrique Martínez shows that the IO should be performed sterile with previous hand washing. To avoid infections, it is convenient to surround the intraosseous needle with a sterile dressing, disinfect the area every 4–6 h, and replace the dressing when it is dirty or wet. Additionally, he shows that the removal of the intraosseous device must be performed by a qualified professional [13]. Although IO is reserved for emergencies and critical care conditions, there are documented cases of quadriplegic patients for many years due to progressive spinal muscular atrophy and refractory disease. IO access was used for palliative sedation with propofol in a home care setting [23].

Regarding the limitations of the study, although the results obtained are conclusive in response to the objectives of the study, larger samples could yield more conclusive results. The heterogeneity between the studies means that the results found should be taken with caution. Given the paucity of published clinical trials, it is difficult to address and see how these techniques affect patients comprehensively, therefore justifying the conduct of future research.

## 5. Conclusions

With this review, we update the evidence of the theoretical and practical knowledge and strategy for the application of the insertion and proper management of the intraosseous route as an emergency vascular access. We show that the proximal humerus bone is the first option in adults. Through this criterion, we achieve higher infusion rates, less pain on insertion, and greater drug bioavailability. The flow velocity is greater in the proximal tibia bone than in the distal tibia bone. As for the sternal access, this allows us to reach the central venous circulation due to its proximity through the mammary glands. Regarding the technical materials, the EZ IO T.A.L.O.N device is designed to be placed on the sternum and is the one that presents the best efficiency in application time.

## Data Availability

Not applicable.

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
