# Peer review of "Clinical Management of Intraosseous Access in Adults in Critical Situations for Health Professionals"

_healthcare, 2022, doi:10.3390/healthcare10020367_

Round 1
Reviewer 1 Report
Thank you for submitting this valuable work reviewing the safety and efficacy of IO use in clinical management for fluid resuscitation or other health care issues. This manuscript is a narrative review of the literature.
Title: suggest adding Narrative Review to guide the reader in the scope of the paper
Abstract: appropriate
Introduction: Adequate review of the literature regarding general IO use/application and various devices available. Page 2, lines 89-92 Need to clarify the intent of the paragraph.
Materials and Methods: Did you register the narrative review protocol (like with Open Science Forum)? Databases: Google Scholar is not a robust searchable platform and often contains articles that were submitted to non-peer reviewed journals (predatory journals) so I would indicate if you could confirm that all manuscripts that meet inclusion criteria were from reliable peer-reviewed journals. Did you perform a quality assessment and if so, name it and describe it. Need to describe the process you and co-authors implemented to review the titles (how many people did this) and abstracts, reach consensus, and then how you performed the extractions of information. Did you use an extraction platform (like Covidence) or construct your own template?
Results: Table 1 – what does the term valid mean? I assume it relates to some component of the inclusion/exclusion criteria but “valid” is not used in any of those listed. If valid refers to a quality assessment, indicate the parameters and overall results. PICO should be listed in Materials & Methods as it is used to direct your literature search and use of key words. You could consider constructing a PRISMA-like diagram to illustrate the table reflecting your methodological flow for the review. I do not understand the purpose of the paragraph following Table 1 (bottom page 4 to top page 5, lines 177-194). Was this your extraction questions of the reporting of articles that answered the extraction questions? This needs revising significantly to help the reader understand the intent as I believe it was what you found from the 18 articles – such as “Four articles described …”
Discussion: This section would benefit from revision as you should offer a synthesis of your review of outcomes; it appears that you are reporting the outcomes but not providing clinicians direction of how to implement your findings into practice. I am not familiar with referring to citations by last name first initial but by last name et al. Line 196 has one author listed but 2 articles cited on line 197.
Conclusion: Adequate. Delete last sentence, line 269, as it is not appropriate in conclusions.
Author Response
We appreciate your assessment of the manuscript, as well as the major and minor suggestions considered to enrich it.
As for the main points:
1) Title: by reflecting it in the abstract, we consider that it was sufficient for the reader's guide. Even so, we have to consider it to try to modify it.
2) Introduction: a modification of the suggested paragraphs has been made.
3) Materials and methods: We only took 2 articles from the Google Scholar database and checked the strict compliance with the inclusion criteria of the article, as well as the methodological quality.
Regarding the evaluation of the quality of the articles, we have used the tools of “PEDro Evaluation Scale” and The Scottish Intercollegiate Guidelines Network. We have incorporated the information into the methodology. The analysis table of the articles, of own elaboration, is also part of the article as an appendix.
4) Results: We use the valid term to clarify the final article included in the review. Regarding the incorporation of the word PICO in the methodology or results section, there is discrepancy between reviewers of different journals, therefore we leave it to their decision in which section to place it. The PRISMA-type diagram has been constructed to illustrate the table that reflects its methodological flow for the review.
5) Discussion: has been fully revised.
6) Conclusion: has been modified.
We especially appreciate your excellent review of the article. We have proceeded to review it completely and in detail, especially the translation. We hope it is of your consideration.
Very thankful.
Reviewer 2 Report
The article entitled " Clinical management of intraosseous access in adults in critical situations for health professionals" revealed that intraosseous infusion is an effective and safe technique, which increases patient survival. The study seems to investigate a topic about knowledge - how to handle the different intraosseous devices in situations in which it is not possible to achieve immediate peripheral venous access.
However, I have some suggestion which should be addressed in the final version.
First of all, I don’t understand why the authors called this review as a narrative review. Typical narrative reviews have no predetermined research question or specified search strategy, only a topic of interest. The authors showed different way about their paper, like they try to prepare the systematic review, which obviously is not. Thus, I suggest to decide what kind of paper the authors want to present. The results showed at the manuscript could be useful to prepare the systematic review according for example to PRISMA (Page M J, McKenzie J E, Bossuyt P M, Boutron I, Hoffmann T C, Mulrow C D et al. The PRISMA 2020 statement: an updated guideline for reporting systematic reviews BMJ 2021; 372 :n71 doi:10.1136/bmj.n71). The authors claimed that the paper was constructed following the PICO format. Indeed, it will be useful for systematic review. The authors should present the database searching on flow diagram and show the process of the searching. Description about any methods used to assess risk of bias due to missing results in a synthesis should be also added. It could be interesting to show the data from these 18 papers in the table, or maybe in another way which can present to the readers the specific description of the effectiveness and comparison between devices. Moreover, the limitations of the study should be addressed in the discussion section. In sum, the major revision is needed. In my opinion this article should be rewrite and the data should be present as a typical narrative review or change to systematic review which could will be more valuable for readers. Moreover, the home-message should be more substantial.
Author Response
Dear reviewer,
We appreciate your assessment of the manuscript and we highly value major and minor suggestions to enrich it.
Regarding the main points:
1) In our review we introduce a predetermined research question to a specific search strategy with a topic of interest. The article is a systematic review. We have made the modification in the abstract and in the material and methods section.
2) We incorporate as an appendix at the end of the article the database used to justify the revision.
3) In the same way, we incorporate a flowchart that details the search carried out
4) We added a paragraph on the limitations of the study at the end of the discussion section.
5) We carry out a new revision of the translation to the department that was commissioned.
We especially appreciate your excellent review of the article. We have proceeded to review it completely and in detail. We hope it is of your consideration.
Very thankful.
Reviewer 3 Report
I congratulate the authors for their work. I suggest incorporating more evidence into the discussion and enriching the conclusions.
Author Response
Dear reviewer,
We appreciate your assessment of the manuscript and greatly appreciate major and minor suggestions to enrich it.
As for the main points:
1) We have revised the discussion. We have also modified the conclusions considering that they are much more enriched.
2) We have also reviewed the translation with the corresponding department.
We especially appreciate your excellent review of the article. We have proceeded to review it completely and in detail. We hope it is of your consideration.
Very thankful.
Reviewer 4 Report
Dear authors,
The methodology of intraosseous access is explained in detail, and I thought it was very easy to understand.
On the other hand, there are many incomplete points as a review paper.
I think there are too many non-English papers. Quoting non-English papers is no problem, however there are many published English papers on intraosseous access, so I think you should cite them.
Isn't dieckman dieckmann?
I think the "objective" is abstract. Most of the "Introduction" explain anatomy and how to operate intraosseous access.
It does not explain how intraosseous access is used clinically, its effectiveness, and issues.
Therefore, the direction of the review is unknown, and as a result, the "objective" is abstract.
The keywords include "diabetes, foot, ulcers, prevention, self-care and education". However, it is not possible to understand why these was selected.
It might not be appropriate that the results include review paper (Reference No.3).
This paper summarized previous studies about intraosseous access and introduce how to use intraosseous access.
What you are trying to do with is a kind of similar with this paper.
If you cite it, I think you should refer at least the original paper, not this paper.
18 out of 117 seems to be valid, however, I couldn't understand the criteria.
There are only a few lines in the result. Rather, the content of this result should be categorized and summarized in a table.
What written in the "discussion" is the introductions of reviewed papers. I think it is "result" rather than "discussion".
As for the conclusion, the results and discussion are not sufficiently written, so it is not possible to judge whether this conclusion is valid.
Author Response
Dear reviewer,
We appreciate your assessment of the manuscript and greatly appreciate major and minor suggestions to enrich it.
As for the main points:
1) Articles in both English and non-English are reviewed and updated. We appreciate the consideration.
2) Isn't dieckman dieckmann? There is published literature of both forms. Still, let's consider "Dieckmann".
3) The objective of the revision of the modified to: The general objective of this work is to update the theoretical-practical strategy of the insertion and proper management of the intraosseous route as emergency vascular access for health professionals.
4) Clinical intraosseous access and its efficacy are updated in the introduction.
5) We appreciate the appreciation of Reference Nº3, but we consider that it is an important article due to its practical justification of the content.
6) A table is inserted as appendix 1, also as a recommendation from another reviewer, which summarizes the different articles used.
7) Both the discussion and the conclusions are partially updated and modified.
We especially appreciate your excellent review of the article. We have proceeded to review it completely and in detail. We hope it is of your consideration.
Very thankful.
Round 2
Reviewer 1 Report
Thank you for your revisions. You have addressed the concerns and significantly improved the quality of the submission.
Reviewer 2 Report
The Authors have followed my comments
Reviewer 4 Report
Dear authors,
The manuscript was significantly revised and I was able to understand the significance of this paper. There may be room for elaboration from an academic point of view, but I think it is sufficient for the content to be delivered to clinicians. I hope this paper will lead to better medical care.